# Statistical Characterization of the Magnetic Field in Space during Magnetic Storms

**Shi-Han Wang** [1,2,†], **Lei Li** [1,2,†] , **Tao Chen** [1,*], **Shuo Ti** [1], **Chun-Lin Cai** [1], **Wen Li** [1] and **Jing Luo** [1]

1   State Key Laboratory of Space Weather, National Space Science Center, Chinese Academy of Sciences, Beijing 100190, China
2   School of Earth and Planetary, University of Chinese Academy of Sciences, Beijing 100049, China
*   Correspondence: tchen@nssc.ac; Tel.: +86-139-1042-1558
†   These authors contributed equally to this work.

**Abstract:** Magnetic storms are an important type of space weather and are usually caused by large streams of charged elementary particles (ions, for example) generated during solar wind production. The occurrence of magnetic storms can pose a threat to the internal electronics of satellites, communication, navigation, remote sensing, etc. Additionally, ground-based electrical facilities may be impacted. In this paper, we focus on the statistical characteristics of the space channel during the occurrence of magnetic storms. By analyzing the observed data for each component of the magnetic field during a magnetic storm and applying the relevant cognitive radio theory, we obtain the probability density function, autocorrelation function, and power spectrum of the magnitude of each component of the magnetic field. The results show that the probability density of the magnitude of each component of the magnetic field gradually deviates from the Gaussian distribution as the Magnetic storm ring current index (Dst index) increases during a magnetic storm, and the autocorrelation function exhibits nonstationary characteristics, which further leads to the time-varying characteristics of the power spectrum.

**Keywords:** magnetic storms; probability density estimation; autocorrelation function estimation; power spectrum estimation

## 1. Magnetic Storms

When coronal mass ejections and high-speed solar winds caused by solar activity impact the magnetosphere, the geomagnetic field will change drastically in a short period of time, thus causing magnetic storms [1–3]. The generation of magnetic storms will affect satellites in orbit and lead to satellite crashes, such as the one that occurred on 8 February 2022 at SpaceX, where 40 of a chain of 49 satellites were affected by magnetic storms and crashed [4]. Moreover, for the communication system, if the solar storm is too strong, it will damage the ground communication system and severely distort the signal, affecting the communication quality or even causing communication interruptions [5,6]. The violent geomagnetic perturbations can lead to navigation and positioning problems [7]. The key factor that leads to the generation of magnetic storms is the presence of a strong southward component of the interplanetary magnetic field over a long duration. Magnetic storms are accompanied by violent perturbations of the near-Earth space environment and are among the most important events in space environment forecasting. The Magnetic storm ring current index (Dst index) is the result of averaging the hourly geomagnetic horizontal perturbation values of four geomagnetic stations near the Earth's equator at approximately uniform longitude intervals with a certain weighting. Notably, values of Dstmin $\geq -19$ nT, $-20$ nT $\geq$ Dstmin $\geq -49$ nT, $-50$ nT $\geq$ Dstmin $\geq -99$ nT, $-100$ nT $\geq$ Dstmin $\geq -249$ nT, and $-250$ nT $\geq$ Dstmin represent magnetic calm periods, small magnetic storms, moderate magnetic storms, strong magnetic storms, and super magnetic storms, respectively. The

magnetic storms selected in the paper are medium-large and strong magnetic storms that occurred from 2016 to 2021 with $-80$ nT $\geq$ Dstmin.

In this paper, six magnetic storm events were selected for analysis for storms occurring from 00:00 on 6 March 2016, to 23:00 on 7 March 2016; 00:00 on 13 October 2016, to 23:00 on 15 October 2016; 01:00 on 27 May 2017, to 23:00 on 28 May 2017; 0:00 on 8 September 2017, to 23:00 on 9 September 2017; 00:00 on 27 September 2020, to 23:00 on 28 September 2020; and 00:00 on 4 November 2021, to 23:00 on 5 November 2021. Figure 1 shows the magnetic field components Bx, By, and Bz for these time periods and the Dst index.

In many of the previous studies on magnetic storms, authors used to focus on solar wind-magneto sphere interaction during magnetic storms, such as large-scale magnetopause reconnection [8–10], the time evolution of Dst [11], the seasonal and solar-cycle distribution of storms [12], as well as relationships of storms and substorms [13–15]. However, in the field of signal processing, it is the statistical characteristics of the environment, rather than the time distribution and interaction of the magnetic storms and other factors, that determine the feature of the channels, which do have an effect on the signals.

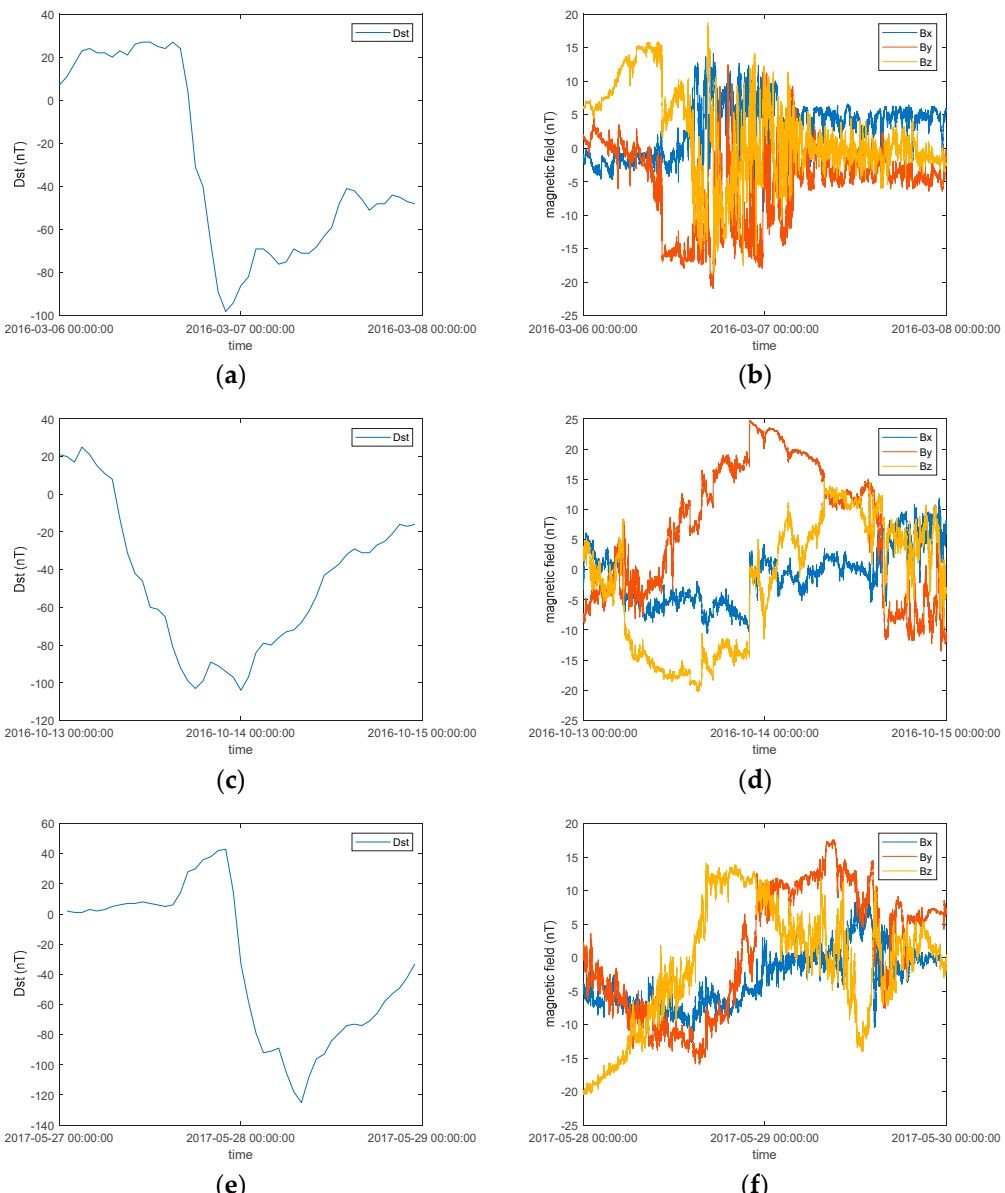

**Figure 1.** *Cont.*

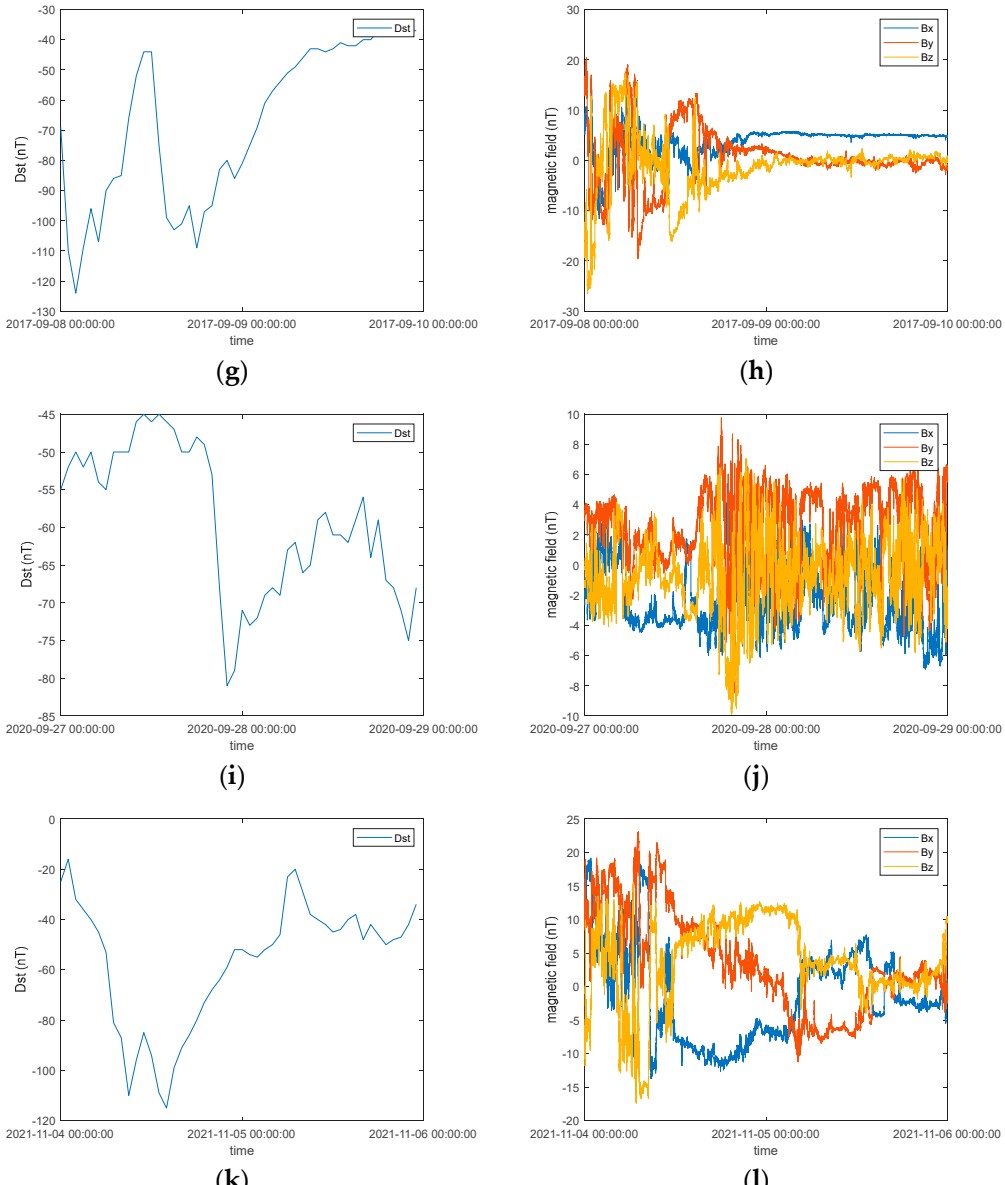

**Figure 1.** Time-domain plot for Dst and components of the magnetic field. (**a**) Dst from 0:00 on 6 March 2016, to 23:00 on 7 March 2016; (**b**) Components of the magnetic field from 0:00 on 6 March 2016, to 23:00 on 7 March 2016; (**c**) Dst from 0:00 on 13 October 2016, to 23:00 on 14 October 2016; (**d**) Components of the magnetic field from 0:00 on 13 October 2016, to 23:00 on 14 October 2016; (**e**) Dst from 0:00 on 27 May 2017, to 23:00 on 28 May 2017; (**f**) components of the magnetic field from 0:00 on 27 May 2017, to 23:00 on 28 May 2017; (**g**) components of the magnetic field from 0:00 on 27 May 2017, to 23:00 on 28 May 2017; (**h**) components of the magnetic field from 00:00 on 8 September 2017, to 23:00 on 9 September 2017; (**i**) Dst from 00:00 on 27 September 2020, to 23:00 on 28 September 2020; (**j**) components of the magnetic field from 00:00 on 27 September 2020, to 23:00 on 28 September 2020; (**k**) Dst from 00:00 4 November 2021, to 23:00 4 November 2021; (**l**) omponents of the magnetic field from 00:00 on 4 November 2021, to 23:00 on 5 November 2021.

Therefore, this paper focuses on the statistical characteristics of the space channel during the occurrence of magnetic storms, which are of great importance to signal analysis for Spaceborne equipment. By analyzing the observed data for each component of the magnetic field for six magnetic storms from 2016–2021, the statistical characteristics of the space magnetic field during magnetic storms, including probability density function, autocorrelation function, and power spectrum, could be obtained. The Dst index used

in the paper is obtained from the rapid-view data released in real-time by WDC Kyoto, the Tokyo station of the World Data System (https://wdc.kugi.kyoto-u.ac.jp/index.html (accessed on 10 July 2022)). Magnetic storm forecasting for alerts at the L1 point can also be used by monitoring the solar wind in real-time using various satellites, such as the ACE satellites. The ACE satellites were launched by NASA as part of the Explorer program on 25 August 1997 and are currently in the vicinity of L1 Liza in orbit. The point is located on a straight line between the Sun and Earth, approximately 1.5 million kilometers from Earth, and is used to study the solar and outer space regions, including solar wind energy particles, interplanetary and interstellar media, and types of galactic matter. Additionally, the real-time information transmitted by ACE satellites is used by the Space Weather Forecasting Center to improve the reliability of solar storm forecasts and warnings. The data for magnetic field components (Bx, By, and Bz) in the text are provided by the ACE satellites.

In Figure 1, the horizontal axis represents the date, and the vertical axis represents the Dst index and magnetic field strength, both in nT. The plots show that when the Dst index decreases significantly, indicating the occurrence of a magnetic storm, all three components of the magnetic field detected by the ACE satellites display positive and negative jitter, indicating that the magnetic field at the detection point fluctuates greatly during a storm. There is a clear temporal relationship between the fluctuations in the magnetic field components and the decrease in the Dst index, with the variation in the Dst index appearing several hours later than the large fluctuations in the magnetic field components, indicating that the magnetic storm first passes through the Lagrangian point and then through the magnetosphere and ionosphere, consequently affecting the magnetic field components near the ground.

## 2. Probability Density Estimates of the Magnitudes of the Components of the Magnetic Field during a Magnetic Storm

Probability density is one of the important characteristics of stochastic processes, especially for digital communication, radar, and navigation systems. The probability density of a signal is directly related to the signal information entropy, correlation function, correlation matrix, and other parameters [16–18], thus affecting the output performance of radar, navigation, and communication systems. For example, classification, denoising, and target detection for synthetic aperture radar imaging are dependent on the amplitude distribution of the received signal [19–21]. In wireless communication systems, the statistical properties of the channels will have an impact on parameter estimation for the target signal [22]. Additionally, the robustness and accuracy of navigation systems are highly dependent on the probability density function [23,24]. Therefore, in this paper, probability density estimation is first performed for the observed data.

Since a priori knowledge is available only from observed data and the mathematical form of probability density functions of the components of the magnetic field during a magnetic storm cannot be obtained directly, a nonparametric estimation method is needed. The Parzen-Rosenblatt probability density estimation method was proposed by Rosenblatt et al. [25,26] for this problem. For observations $x_1, x_2, \ldots, x_N$, the probability density of the random variable $X$ can be estimated using the available data through Equation (1):

$$\hat{f}_X(x) = \frac{1}{Nh} \sum_{i=1}^{N} k\left(\frac{x - x_i}{h}\right) \tag{1}$$

where $f$ with a "^" above is the estimated probability density function, $k(x)$ is the kernel function, and $h$ is the smoothing parameter, also called the bandwidth, and is used to control the value of the kernel function. The probability density estimation approach in

Equation (1) is asymptotically unbiased [27]; i.e., the probability density estimate converges to the true probability density when the sample size is sufficiently large. Namely

$$\lim_{N \to \infty} E\left( \hat{f}_X(x) \right) = f_X(x) \tag{2}$$

where $f$ denotes the actual probability density function, and $E$ denotes the mathematical expectation operator. There are a variety of kernel functions for probability density function estimation. In this paper, we chose the Gaussian kernel function, the mathematical formula of which is shown in Equation (3).

$$k(x) = \frac{1}{\sqrt{2\pi}} \exp\left( -\frac{x^2}{2} \right) \tag{3}$$

where $k$ is the kernel, as depicted in Equation (1). The sampled data are then substituted into Equations (1) and (3). The probability density of the amplitude of each component of the magnetic field could be obtained for the six magnetic storm events, as shown in Figure 2.

A comparison of the Dst results in Figure 1 reveals that the Dst levels on 8 September 2017 and 27 September 2020 are lower than those on the other four days, and the probability density of the magnetic field amplitude generally follows a Gaussian distribution; however, the probability density of the magnetic field amplitude deviates from the Gaussian distribution when the Dst level is higher, which might result from the energy injection during a magnetic storm.

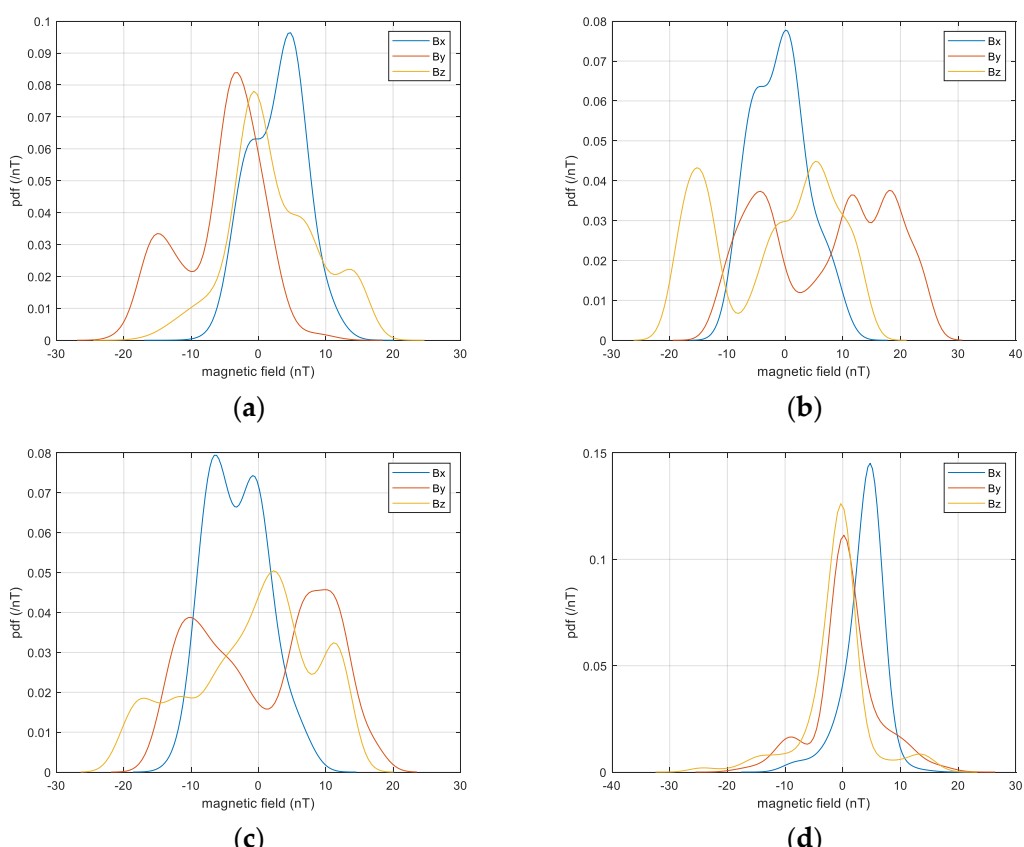

**Figure 2.** *Cont.*

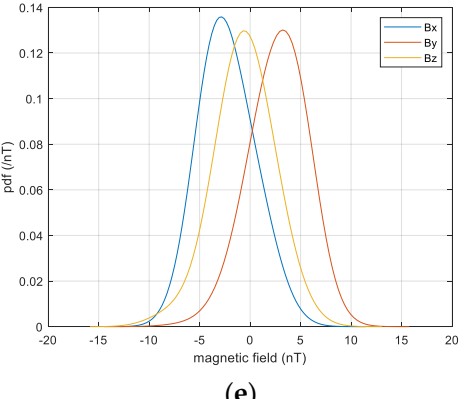 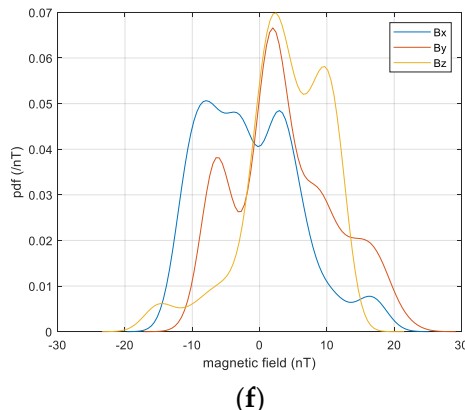

**(e)**　　　　　　　　　　　　　**(f)**

**Figure 2.** Probability densities of the magnitude of each component of the magnetic field. (**a**) 0:00 on 6 March 2016, to 23:00 on 7 March 2016; (**b**) 0:00 on 13 October 2016, to 23:00 on 14 October 2016; (**c**) 0:00 on 27 May 2017, to 23:00 on 28 May 2017; (**d**) 00:00 on 8 September 2017, to 23:00 on 9 September 2017; (**e**) 0:00 on 27 September 2020, to 23:00 on 28 September 2020; (**f**) 00:00 on 4 November 2021, to 23:00 on 5 November 2021.

In fact, according to the central limit theorem, when the sample size is large enough, the limit of many distributions is Gaussian. However, when a magnetic storm occurs, the energy injection might bring some disturbance, and some of the values of observed data might deviate from that in a calm space environment, which would probably violet the necessary condition for the central limit theorem and have some impacts on the information entropy of the magnetic field, and thus result in the deviation from Gaussian distribution.

## 3. Autocorrelation Function for Each Component of the Magnetic Field during a Magnetic Storm

The autocorrelation function is a measure of the similarity between a signal and a delayed signal and is based on an inverse Fourier transform of the power spectrum. The estimation of the autocorrelation function directly affects the accuracy of the estimation of the power spectrum. It thus has an impact on the performance of radar, navigation, and communication systems. For example, when detecting the position and velocity of a target using Doppler radar, the autocorrelation function can be used to directly determine the fuzziness of the signal and, thus, the resolution of the estimates of distance and velocity [28]. When performing array processing, the autocorrelation function determines the properties of the autocorrelation matrix, which has an impact on the performance of adaptive beamforming and direction estimation algorithms [29–32]. In wireless communications, autocorrelation estimation is an effective method for estimating fundamental tones in noise and is also the key to blind signal separation and channel equalization techniques for distributed MIMO communication systems [33,34].

For stationary stochastic processes, the autocorrelation function is only related to the time interval between two samples and is independent of the initial observation moment, i.e.,

$$R(t, t + \tau) = E(x(t)x(t + \tau)) = R(\tau) \tag{4}$$

However, we cannot guarantee that the magnetic fields during a magnetic storm are always stationary processes; therefore, when estimating the autocorrelation function, Equation (4) needs to be applied based on the time delay unit at different initial observation moments. In reality, the mathematical expectation operator in Equation (4) cannot be computed directly through averaging. The specific procedure is implemented by replacing

set averaging with time averaging; i.e., the autocorrelation function is estimated based on convolution operations. In this case, Equation (4) is transformed into

$$R(\tau) = \frac{1}{T}\int_0^T x(t)x(t+\tau)\mathrm{d}t \tag{5}$$

Any value in the sampled data set can be selected as an initial observation value, and the corresponding sampling moment is the initial observation moment. The other samples constitute the sampled values after certain time delays, and the corresponding moments are the instantaneous delays. The above operation is performed by traversing each sampling moment to obtain the three-dimensional autocorrelation function $R(t,\tau)$.

By substituting the sampled data, as shown in Figure 1, into the calculation, the three-dimensional autocorrelation function for each component of the magnetic field can be obtained for each magnetic storm period, as shown in Figure 3.

Figure 3 shows that during a magnetic storm, the autocorrelation function of each component of the magnetic field is non-stationary and can be expressed as a binary function of the time delay and the initial observation moment. In other words, for the same time delay, the autocorrelation function varies at different initial observation moments.

The non-stationary property of the auto-correlation functions of the magnetic fields during magnetic storms might result from the disturbance, which might lead to some electromagnetic pulses whose second moment is non-stationary. In addition, the non-Gaussian property of the probability density function of the magnetic fields might also result in non-stationarity.

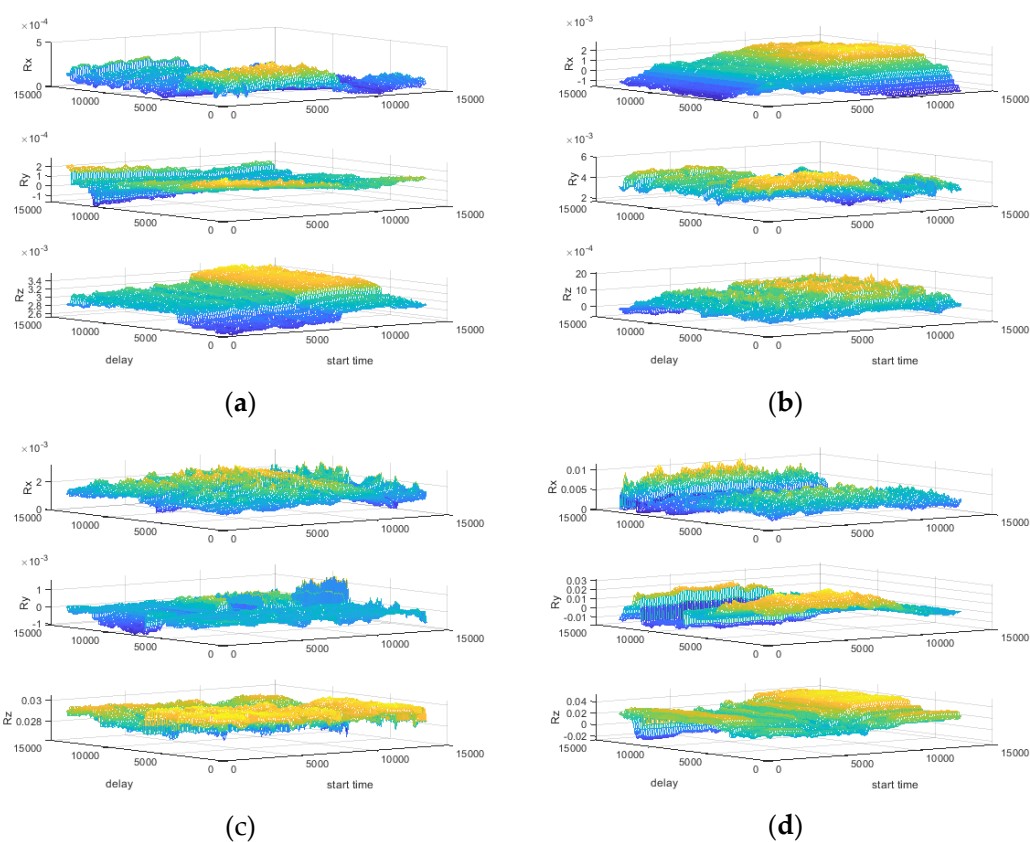

**Figure 3.** *Cont.*

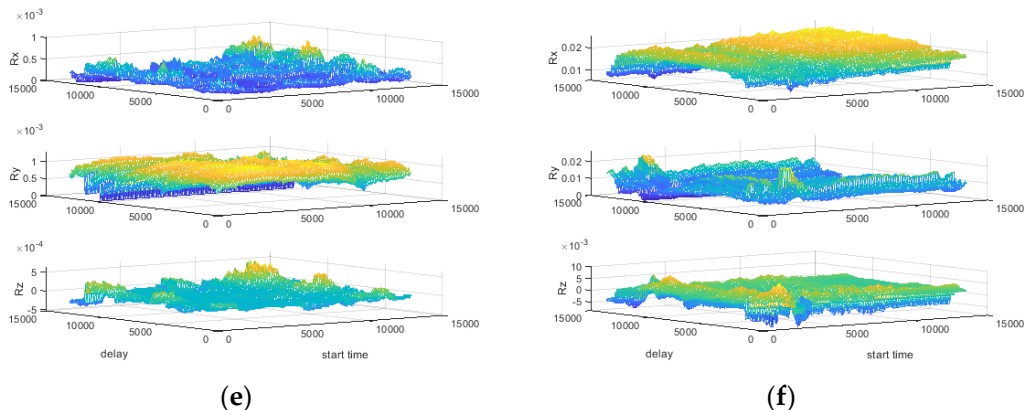

(**e**)　　　　　　　　　　　　　　　　　　　　　　　　　(**f**)

**Figure 3.** The magnitude of each component of the magnetic field. (**a**) 0:00 on 6 March 2016, to 23:00 on 7 March 2016; (**b**) 0:00 on 13 October 2016, to 23:00 on 14 October 2016; (**c**) 0:00 on 27 May 2017, to 23:00 on 28 May 2017; (**d**) 00:00 on 8 September 2017, to 23:00 on 9 September 2017; (**e**) 0:00 on 27 September 2020, to 23:00 on 28 September 2020; (**f**) 00:00 on 4 November 2021, to 23:00 on 5 November 2021.

## 4. Power Spectrum of Each Component of the Magnetic Field during a Magnetic Storm

The power spectrum is an important numerical feature of a stochastic process. It is the Fourier transform of the autocorrelation function, indicating the amount of power contributed by each frequency component of the stochastic process. Power spectrum estimation is widely used in cognitive radio analysis, spectrum multiplexing, transmit power control, and interference suppression [35–37]. In wireless communication systems, power spectrum measurements are crucial for techniques such as delay estimation, channel equalization, and speech enhancement [38–40].

For a stationary stochastic process, the power spectrum is the Fourier transform of the autocorrelation function:

$$S(\omega) = \int_{-\infty}^{+\infty} R(\tau)e^{-j\omega\tau}d\tau \tag{6}$$

The analysis in Section 3 shows that the magnetic field components exhibit non-stationary properties under magnetic storm conditions, so it is necessary to introduce a time variable in addition to frequency; i.e., for non-stationary processes, the power spectrum is a binary function of both time and frequency, indicating the magnitude of the power contributed by a frequency component at a given observation time. The result is referred to as the dynamic power spectrum [41]. The dynamic power spectrum and the observed time series satisfy

$$\begin{aligned} \int_{-\infty}^{+\infty} S(t,\omega)d\omega = |x(t)|^2 \\ \int_{-\infty}^{+\infty} S(t,\omega)dt = |X(\omega)|^2 \end{aligned} \tag{7}$$

where $S(t,\omega)$ is the dynamic power spectrum, $x(t)$ is the observed value, and $X(\omega)$ is the Fourier transform of $x(t)$.

According to the conditions in (7), the dynamic power spectrum can be expressed as described in [41] for a non-stationary stochastic process:

$$S(t,\omega) = \int_{-\infty}^{+\infty} E(x(t)x(t+\tau))e^{-j\omega\tau}d\tau = \int_{-\infty}^{+\infty} R(t,\tau)e^{-j\omega\tau}d\tau \tag{8}$$

By substituting the binary autocorrelation function obtained in Section 3 into the above equation, the dynamic power spectrum of each component of the magnetic field during the six magnetic storms can be obtained, as shown in Figure 4.

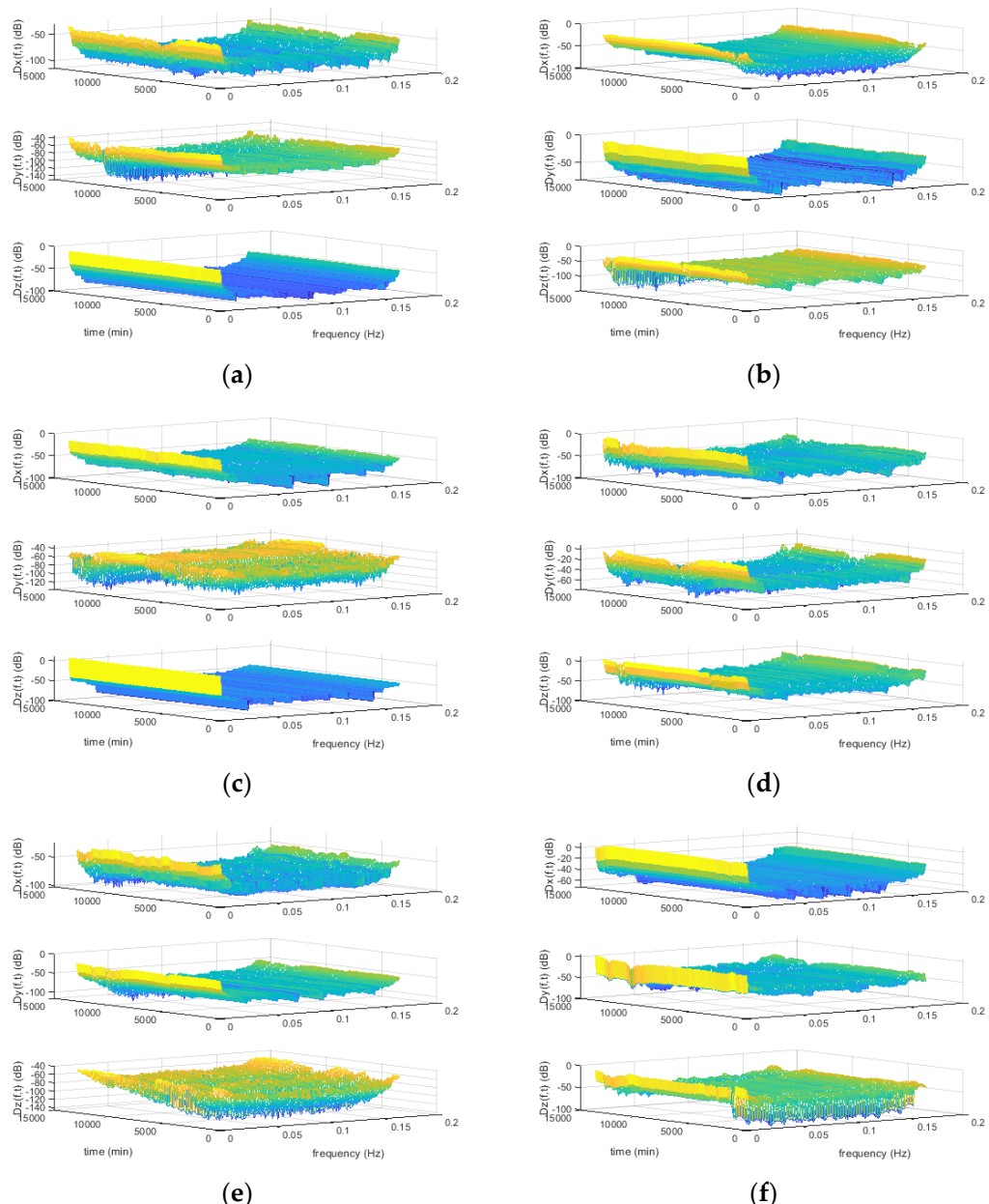

**Figure 4.** Dynamic power spectrum of the magnitude of each component of the magnetic field. (**a**) 0:00 on 6 March 2016, to 23:00 on 7 March 2016; (**b**) 0:00 on 13 October 2016, to 23:00 on 14 October 2016; (**c**) 0:00 on 27 May 2017, to 23:00 on 28 May 2017; (**d**) 00:00 on 8 September 2017, to 23:00 on 9 September 2017; (**e**) 0:00 on 27 September 2020, to 23:00 on 28 September 2020; (**f**) 00:00 on 4 November 2021, to 23:00 on 5 November 2021.

For a nonstationary magnetic field during magnetic storms, the dynamic power spectrum in each observation period is mainly concentrated in the very-low-frequency band, with an attenuation of up to 100 dB or more as the frequency rises. This result confirms the conclusion that the fluctuations caused by magnetic storm excitation occur in the very-low-frequency band [42,43].

## 5. Results and Discussions

The results show that the probability distribution of the magnitude of each component of the magnetic field is a near-Gaussian distribution when the Dst index is low; however, this distribution deviates from the Gaussian distribution when the Dst index is high, i.e., when magnetic storms are relatively strong. This is probably because the energy

injection during a magnetic storm brought some disturbances. Thus some of the values of observed data might deviate from that in a calm space environment, which would probably violate the necessary condition for the central limit theorem and have some impacts on the information entropy of the magnetic field.

The auto-correlation function of each component of the magnetic field is non-stationary during a magnetic storm. The auto-correlation function is simultaneously a binary function related to the initial observation moment and the time delay between two observations. This property might result from both the electromagnetic pulses with a non-stationary second moment because of the disturbances during the magnetic storm and the non-Gaussian property of the probability density function of the magnetic fields.

The dynamic power spectrum analysis shows that for each observation moment, the power spectrum is mainly concentrated in the very-low-frequency band. The result is similar to that of previous research, as shown in [34,35].

**Author Contributions:** T.C. designed the project and wrote the paper; S.-H.W. collected, processed, and analyzed the data; S.-H.W. prepared the original draft with contributions from all authors; L.L., S.T., C.-L.C., W.L. and J.L. were responsible for discussion and revisions. All authors have read and agreed to the published version of the manuscript.

**Funding:** This study is supported by the Strategic Pioneer Program on Space Science, Chinese Academy of Sciences (Grant Nos. XDA17010301, XDA17040505, XDA15052500, and XDA15350201), the National Natural Science Foundation of China (Grant Nos. 41731070 and 41931073), the Specialized Research Fund for State Key Laboratories, the CAS-NSSC-135 project and the NSSC Director Fund (Grant No. E0PD41A11S).

**Institutional Review Board Statement:** Not applicable.

**Informed Consent Statement:** Not applicable.

**Data Availability Statement:** The datasets used or analyzed in the present work are available from the corresponding author upon reasonable request.

**Acknowledgments:** The authors thank swx.sinp.msu.ru for the data.

**Conflicts of Interest:** The authors declare no conflict of interest.

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
