# Peer review of "Statistical Characterization of the Magnetic Field in Space during Magnetic Storms"

_atmosphere, doi:10.3390/atmos13101578_

Round 1

Reviewer 1 Report (Previous Reviewer 1)

I propose that the revised version can be published in the present form.

Reviewer 2 Report (Previous Reviewer 3)

I satisfy the revisions by authors.

This manuscript is a resubmission of an earlier submission. The following is a list of the peer review reports and author responses from that submission.

Round 1

Reviewer 2 Report

Manuscript: atmosphere-1912200

“Statistical characterization of the magnetic field in space during magnetic storms” by Shihan Wang et al.

Magnetic storms are an important type of space weather. In this paper, the authors analyzed different components of the IMF magnetic field during several magnetic storms and obtained the probability density function, autocorrelation function, and power spectrum characteristics of the magnitude of each component of the magnetic field. Although this work is interesting, the authors only briefly introduce these results without necessary analysis and discussion. There are many typographical and editing errors in this paper. For example, there are two second sections and two fifth sections, and the first paragraph of section 3 is also strange. Hence, I do not think this paper can be published in atmosphere. 

Reviewer 3 Report

The submission by Shihan Wang et al. focuses on the characterization of the magnetic field during magnetic storms, and obtains the probability density function, autocorrelation function, and power spectrum characteristics of the magnitude of each component. The results show that the probability density gradually deviates from the Gaussian distribution as the Dst index increases, and the autocorrelation function exhibits nonstationary characteristics, further, leading to the time-varying characteristics of the power spectrum. The result should be useful in space weather community. I would like to recommend to publish on atmosphere after some major revisions as below.

1.  In section 1, some methods for the analysis of magnetic storm should be given sufficiently.

2.     Line 39, the abbreviation ‘Dst’ should be given in detain.

3.     Line 55, it may be better to delete “summarizing”.

4.     In function (2), what is the E?, which should be clarified.

5.     The three functions (1), (2) and (3) should be clarified.

6.     In Figure 1, the Dst index and three components of field, measured in September 2020 and November, 2017, have not been given. Right? That is, Figure 1 is only some examples measured?

7.     From line 183 to line195, it seems not to understand, which needs rewriting.